# Repurposing Historic Drugs for Neutrophil-Mediated Inflammation in Skin Disorders

**DOI:** 10.3390/biom14121515

**Published:** 2024-11-27

**Authors:** Ludovica Franceschin, Alessia Guidotti, Roberto Mazzetto, Jacopo Tartaglia, Christian Ciolfi, Mauro Alaibac, Alvise Sernicola

**Affiliations:** Dermatology Unit, Department of Medicine (DIMED), University of Padua, 35121 Padova, Italy; ludovica.franceschin@studenti.unipd.it (L.F.); alessia.guidotti@studenti.unipd.it (A.G.); jacopo.tartaglia@studenti.unipd.it (J.T.); chriciolfi96@gmail.com (C.C.); mauro.alaibac@unipd.it (M.A.)

**Keywords:** dapsone, colchicine, tetracycline, neutrophils, autoimmune bullous disorders, psoriasis, pyoderma gangrenosum, hidradenitis suppurativa

## Abstract

Neutrophil-mediated inflammation is a key feature of immune-mediated chronic skin disorders, but the mechanistic understanding of neutrophil involvement in these conditions remains incomplete. Dapsone, colchicine, and tetracyclines are established drugs within the dermatologist’s therapeutic armamentarium that are credited with potent anti-neutrophilic effects. Anti-neutrophilic drugs have established themselves as versatile agents in the treatment of a wide range of dermatological conditions. Some of these agents are approved for the management of specific dermatologic conditions, but most of their current uses are off-label and only supported by isolated reports or case series. Their anti-inflammatory and immunomodulatory properties make them particularly valuable in managing auto-immune bullous diseases, neutrophilic dermatoses, eosinophilic dermatoses, interface dermatitis, and granulomatous diseases that are the focus of this review. By inhibiting inflammatory pathways, reducing cytokine production, and modulating immune responses, they contribute significantly to the treatment and management of these complex skin conditions. Their use continues to evolve as our understanding of these diseases deepens, and they remain a cornerstone of dermatological therapy.

## 1. Introduction: Dermatoses with Neutrophilic Inflammation

Neutrophils are key players in the immune microenvironment associated with cutaneous inflammatory disorders both in acute conditions, where they act as first responders to inflammation, and in chronic disorders, where they provide enzymes and reactive oxygen species that contribute to tissue damage. Dysregulated neutrophil activity is a feature of chronic inflammation that is pertinent to psoriasis and several immune-mediated inflammatory skin conditions.

Repurposing existing therapeutic agents that are effective against neutrophil inflammation is an attractive approach to expanding the indications of known drugs with an established safety profile. Compared to developing new drugs, it provides a cost-effective approach and potentially a quicker translation to clinical use.

This review analyzes current uses of therapeutic agents targeting neutrophil inflammation, examining the existing literature with a focus on dermatologic conditions. To achieve this objective, PubMed was searched from inception to September 2024; papers written in the English language were screened by the authors to retrieve results relevant to the topic of this review. Moreover, the reference list of included papers was analyzed to identify additional results.

### 1.1. Anti-Neutrophilic Drugs

Dapsone, colchicine, and tetracyclines are established drugs within the dermatologist’s therapeutic armamentarium; however, their use in neutrophil-mediated conditions is mostly off-label and still requires validation from robust clinical trials; conversely, our pathogenetic understanding of the mechanisms of neutrophil involvement in these disorders remains incomplete. Further research will also enable us to weigh the benefits of novel indications against the risks of adverse events with prolonged use.

Dapsone is traditionally used as an antibiotic in leprosy and as an anti-inflammatory in dermatitis herpetiformis, owing to its emerging antineutrophilic effects. Colchicine is a historical anti-inflammatory agent used for gout and familial Mediterranean fever. Tetracyclines are a class of broad-spectrum antibiotics with potent anti-inflammatory effects that are approved for the management of acne and rosacea and used off-label in a diverse range of inflammatory skin conditions (Table 1).

Table 2 summarizes the main mechanisms of action and the anti-neutrophilic effects of the investigated drugs.

### 1.2. Autoimmune Bullous Disorders

Bullous diseases feature blisters and bullae on the skin and mucous membranes, driven by autoantibodies [1]. Common types include pemphigus (vulgaris and foliaceus) and pemphigoid (bullous and mucous membrane), with less common forms including epidermolysis bullosa acquisita, dermatitis herpetiformis, and linear IgA bullous dermatosis.

The management of these diseases often involves immunosuppressive therapies, often carrying significant side effects. Bullous pemphigoid (BP), the most common autoimmune blistering skin disease, predominantly affects the elderly and is characterized by intense itching, erythema, and tense blisters caused by autoantibodies attacking the dermal–epidermal junction. Oral prednisolone still represents the standard treatment, though its use in elderly patients can result in significant adverse effects. Super-potent topical corticosteroids provide a safer alternative but may not be practical for all patients.

Oral corticosteroids are also the primary treatment for pemphigus vulgaris, while the use of traditional corticosteroid-sparing agents remains controversial [2]. Currently, rituximab is established as a highly effective first-line adjunctive therapy in pemphigus vulgaris [3].

### 1.3. Neutrophilic Dermatoses

Neutrophilic dermatoses (NDs) are inflammatory skin conditions with neutrophilic infiltrate on histopathology, without infection [4]. Key NDs include IgA pemphigus, Sweet syndrome, pyoderma gangrenosum, and leukocytoclastic vasculitis. Other related conditions are hidradenitis suppurativa and pustular psoriasis.

#### 1.3.1. Sweet Syndrome

Also known as acute febrile neutrophilic dermatosis, Sweet syndrome is characterized by the sudden onset of painful, erythematous plaques and nodules. The pathogenesis of Sweet syndrome remains incompletely understood. However, its frequent association with infections suggests a potential pathogenic role for bacterial antigens or antigen–antibody complexes. Additionally, its co-occurrence with arthritis and Sjögren’s syndrome, along with the rapid response to corticosteroids, indicates the involvement of immune mechanisms. Although no specific guidelines exist for treating Sweet syndrome, systemic corticosteroids are generally the first-line treatment [5].

#### 1.3.2. Leukocytoclastic Vasculitis

Leukocytoclastic vasculitis (LCV) is a common form of small vessel vasculitis (SVV) that most commonly presents clinically as palpable purpura [6]. Papules usually develop over a few hours, either simultaneously or sequentially, with the lower legs and other dependent areas being the primary site of involvement. The lesions frequently merge to form confluent patches and generally resolve spontaneously within 2–3 weeks, leaving post-inflammatory hyperpigmentation as they fade. In severe, intractable, or recurrent skin-limited LCV, treatment generally involves systemic corticosteroids, with or without adjunctive therapies.

#### 1.3.3. Pustular Psoriasis

Pustular psoriasis comprises several distinct clinical entities: palmoplantar pustulosis (PPP), the acrodermatitis continua of Hallopeau (ACH), and generalized pustular psoriasis (GPP) [7,8]. GPP is a potentially life threating condition whose flares are marked by the acute onset of a rapidly disseminating skin eruption characterized by aseptic pustules, along with systemic symptoms such as high fever (up to 40 °C), fatigue, and general malaise [7,9]. ACH is characterized by pustular lesions affecting the extremities of the hands and feet, often leading to progressive nail apparatus destruction and, in severe cases, bone erosions. PPP is the most common localized variant of pustular psoriasis, predominantly affecting women and strongly associated with smoking. The disease typically manifests as chronic aseptic pustular lesions that progress through various stages, from yellowish scales or crusts to brown macular residual lesions. The treatment of pustular psoriasis is challenging as cyclosporine has the highest level of evidence for efficacy in PPP, while evidence for acitretin and methotrexate is weaker. Clinical trials conducted with secukinumab and guselkumab have not demonstrated clinically significant superiority over a placebo. The response to the IL-1 inhibitor anakinra is documented in isolated cases, including ACH. Novel therapies targeting the IL-36 cytokine pathway are a promising addition to the treatment of pustular psoriasis and may provide valuable options for patients who have not achieved sufficient relief with existing therapies [10].

#### 1.3.4. Hidradenitis Suppurativa

Hidradenitis suppurativa (HS) is a chronic, inflammatory, and debilitating skin disease of the hair follicle. It typically appears after puberty and is characterized by painful, deep-seated, inflamed lesions in areas of the body with apocrine glands, most commonly occurring in the axillae, inguinal, and anogenital regions. The central pathogenic event in HS is believed to be the occlusion of the upper part of the hair follicle, which triggers perifollicular lympho-histiocytic inflammation [11,12]. First-line therapy for mild forms involves the use of topical clindamycin, which is effective in treating papulo-pustular lesions but not deeper nodules. Systemic treatment is required when more severe or widely spread lesions are present.

#### 1.3.5. Behçet’s Disease

Behçet’s disease is systemic vasculitis involving both arterial and venous vessels of all sizes, characterized by oral aphthae, painful genital ulcerations, and ocular abnormalities, including keratitis, optic neuritis, and uveitis. The disease has multiple systemic associations that include the involvement of the gastrointestinal, cardiovascular, and central nervous systems, as well as the joints, blood vessels, and lungs. The disease is more prevalent in countries along the ancient Silk Road, extending from Eastern Asia to the Mediterranean basin, and the human lymphocyte antigen (HLA)-Bw51 has been associated with various manifestations of Behçet’s disease.

### 1.4. Eosinophilic Dermatoses

Eosinophilic dermatoses are rare diseases characterized by eosinophil-rich infiltrates and/or eosinophil degranulation with associated blood eosinophilia [13]. Classic types include eosinophilic cellulitis (Wells syndrome), granuloma faciale, and eosinophilic annular erythema. Wells syndrome, also known as eosinophilic cellulitis, is a rare condition with an often unclear etiology. Patients typically present with recurrent cutaneous swellings that resemble cellulitis. Clinical evidence supporting the efficacy of various treatments is largely anecdotal, and oral antimicrobials generally appear ineffective in treating eosinophilic cellulitis. The management of chronic urticaria is also mentioned within this group.

### 1.5. Interface Dermatitis

Interface dermatitis is characterized by the involvement of the dermo-epidermal junction [14]. Major diseases include lichen planus and lupus erythematosus. Lichen planus (LP) is an autoimmune inflammatory skin disorder characterized by the presence of T lymphocytes at the dermal–epidermal junction. Current therapies for cutaneous LP are limited by short-term efficacy, toxicity, and tolerability issues. Severe cases may respond to retinoids and immunosuppressive agents, although chronic use is often necessary. While some patients benefit from psoralen and ultraviolet A (PUVA) therapy, high relapse rates upon discontinuation have been reported, highlighting the need for novel treatments with improved safety profiles.

### 1.6. Granulomatous Diseases

Cutaneous granulomatosis is marked by granulomatous inflammation and macrophage nodules [15]. Non-infectious examples include granuloma annulare, sarcoidosis, and necrobiosis lipoidica.

Granuloma annulare (GA) is a common non-infectious granulomatous disease, with its pathogenesis likely rooted in immunologic mechanisms. Evidence suggests that GA represents a cell-mediated immune response to an unidentified antigen, where granuloma formation by macrophage and histiocytes is mediated through interactions with T-helper lymphocytes [16]. Localized GA often resolves spontaneously within a few months to a few years, whereas GGA can persist for decades. High-potency topical corticosteroids are considered the first-line therapy for GA.

Sarcoidosis, known for its heterogeneous morphological manifestations, is considered one of the “great imitators” in medicine. Physicians treating sarcoidosis must recognize its diverse organ involvement, particularly in the skin, where lesions can be disfiguring, carry prognostic significance, and may be challenging to diagnose. Therapeutic challenges arise from the chronic nature of the disease and the lack of spontaneous regression of lesions, which necessitates balancing the benefits of treatment with long-term tolerability. Oral corticosteroids, which are beneficial in cases of severe visceral involvement, are not warranted for less severe cases due to their associated adverse effects and the frequent relapses observed during dosage tapering. Given the toxicity of long-term steroid use, there has been increasing interest in corticosteroid-sparing agents for sarcoidosis.

Necrobiosis lipoidica (NL) is a rare, chronic granulomatous skin disease, and its treatment remains challenging. Initial therapy often involves topical or intralesional corticosteroids, though multiple therapies have been tried with inconsistent results. Most data on treatment efficacy come from case reports, case series, or small, uncontrolled studies, with a lack of high-quality trials to confirm the effectiveness of individual therapies.

## 2. Dapsone

Dapsone (diaminodiphenyl sulfone, DDS) is an aniline compound in the sulfone drug class, used primarily for infectious diseases like leprosy and to prevent infections in immunosuppressed patients [17]. DDS acts by affecting folic acid metabolism, crucial for DNA synthesis in infectious agents [18]. In addition to its use in infections, DDS serves as an anti-inflammatory medication for chronic dermatologic conditions, thanks to its corticosteroid-sparing properties.

Hepatic metabolism produces monoacetyl-DDS and N-hydroxyl-DDS, which can cause methemoglobinemia, leading to hypoxia and symptoms like dyspnea and headache [19]. Assessing glucose-6-phosphate dehydrogenase (G6PD) levels before starting DDS therapy is essential, as low levels increase the risk of hemolytic anemia [20]. Other side effects include skin rash, malaise, and gastrointestinal issues, occurring in 0.5–3.6% of patients [20,21]. These are reversible upon discontinuation, with treatments like methylene blue and ascorbic acid available for methemoglobinemia [22,23].

### 2.1. Mechanism of Action

Its anti-inflammatory effects include scavenging oxygen radicals, reducing the tumor necrosis factor (TNF), and disrupting lymphocyte and neutrophil function (Table 2) by interfering with their migration processes [19,24]. These diverse anti-inflammatory properties make DDS effective in dermatologic inflammatory conditions with distinct histologic features and cellular infiltrates.

### 2.2. Autoimmune Bullous Disorders

A multicenter study of 53 patients with pemphigus vulgaris showed DDS as an effective and safe adjuvant therapy with moderate corticosteroid-sparing potential compared to azathioprine [25]. DDS is also a first-choice treatment for mild/moderate mucous membrane pemphigoid (MMP), with or without corticosteroids [20]. Two case reports indicate DDS as an effective adjuvant for topical steroid-resistant juvenile MMP.

DDS remains a first-line treatment for pemphigus foliaceus when combined with topical corticosteroids, although a corticosteroid-sparing effect compared to corticosteroids alone was not demonstrated [26].

DDS helps in epidermolysis bullosa acquisita (EBA) by reducing the occurrence of bullae and erosions [27,28]. It has also shown an 81.2% improvement rate in dermatitis herpetiformis patients treated with a gluten-free diet plus DDS [29]. Additionally, DDS, combined with sulphapyridine, has been successfully used for linear IgA bullous dermatosis, where it is recommended as a first-line treatment [30,31,32].

### 2.3. Neutrophilic Dermatoses

Evidence supporting the use of DDS in this group of disorders is provided by several case series and reports in the literature. However, studies with a randomized trial design are lacking.

In cases of Sweet syndrome, Hrin et al. reported that DDS has been effective in 86% of patients, with 57% achieving complete healing [33].

Systemic DDS showed 15.6% complete healing and 81.3% partial improvement in PG [34,35] and may, thus, represent an effective and tolerable adjuvant treatment for this condition.

In the treatment of LCV, DDS has been little used; however, three patients with the common palpable purpura form of the disease, limited to the skin, were successfully treated in the literature. DDS has also been proposed as a first-line therapy for erythema elevatum diutinum, showing rapid improvement and high effectiveness [36]. In urticarial vasculitis, where systemic steroids and adjunctive agents like omalizumab and mycophenolate mofetil are often used, DDS showed 52% “excellent” or “clear” responses [37].

In a case series of treatment-resistant pustular psoriasis, DDS was effective in four out of five patients [38].

Finally, current guidelines reserve DDS as a third-line agent for patients with mild-to-moderate HS [39]. The use of DDS in this context is supported by a study showing that 62.5% of HS patients had at least a 50% reduction in abscesses and nodules after 12 weeks of therapy [40].

### 2.4. Eosinophilic Dermatoses

Wells syndrome lacks a standardized treatment, but corticosteroids and DDS are commonly used drugs [41]. Granuloma faciale shows variable results with DDS, with no consensus on its effectiveness [42]. Eosinophilic annular erythema (EAE) is typically treated with topical or systemic steroids [43]. For patients resistant to both topical corticosteroids and hydroxychloroquine, DDS is a safe and effective option, with some reports showing complete symptom resolution [44,45]. DDS is also a well-tolerated treatment for chronic idiopathic urticaria when previous treatment options fail [46].

### 2.5. Interface Dermatitis

Antimalarials are the primary treatment for cutaneous lupus erythematosus (CLE). DDS is recommended as a second-line treatment for refractory CLE, particularly for bullous lesions. DDS has also been successfully used for lupus erythematosus profundus, with positive outcomes in 11 cases, and a study of 34 CLE patients showed that DDS was effective in over 50% of cases [47].

For LP, first-line treatments are potent topical steroids, intralesional triamcinolone, and systemic corticosteroids [20]. In a study of 33 patients treated with DDS for 16 weeks, 66.5% had complete healing with hyperpigmentation, while 21.5% had a partial response [48].

Recurrent erythema multiforme (REM) is best treated with continuous oral acyclovir, which is effective in reducing recurrences, especially in HSV-induced cases [49]. A study of 13 REM patients treated with dapsone after failing antiviral therapy found that 46.2% achieved a complete response and 38.5% a partial response [50].

### 2.6. Granulomatous Diseases

Generalized granuloma annulare (GGA) can be persistent and is primarily treated with phototherapy and antimalarials in these cases [51]. A review showed that 54% of patients improved with oral DDS, but 57% of these experienced flare-ups after treatment ended [52].

Rheumatoid papules associated with collagen vascular diseases are best treated with corticosteroid injections; however, these may be painful [53]. DDS was effective in managing a patient with a progressive papular eruption associated with rheumatoid arthritis [54].

## 3. Colchicine

Colchicine, an alkaloid extracted from the seeds of the autumn crocus (*Colchicum autumnale*), has been utilized in medicine for millennia [55]. Over time, the clinical applications of colchicine have broadened beyond the original uses for gout and familial Mediterranean fever to include a range of cardiovascular and rheumatological conditions. Recent research has explored the anti-inflammatory effects of colchicine in dermatology and other fields, including its controversial use in treating COVID-19 patients [56].

Colchicine is recognized as a safe and effective treatment for a wide range of cutaneous disorders. Its affordability, efficacy, and favorable safety profile make colchicine a promising treatment option for various dermatological conditions, even in the current era of targeted therapeutics.

### 3.1. Mechanism of Action

Colchicine exerts its effects primarily by inhibiting microtubule polymerization, which interferes with cell division and modulates intracellular signaling pathways (Table 2). This mechanism is particularly impactful on rapidly proliferating cells, such as those in the bone marrow and gastrointestinal tract. Colchicine preferentially accumulates in leukocytes, especially neutrophils, due to their low expression of P-glycoprotein [57] and inhibits neutrophil activation, degranulation, and migration, thereby reducing inflammation [58]. Colchicine also has antifibrotic effects, which may be beneficial in conditions involving fiber deposition, such as scleroderma and cirrhosis [59].

Colchicine demonstrates good oral bioavailability, with absorption occurring primarily in the jejunum and ileum. Peak plasma concentrations are achieved approximately 2 h post-administration, and the drug has a half-life of about 4 h [60]. Its lipophilic nature facilitates effective absorption. Colchicine is metabolized in the liver by cytochrome P450 3A4 (CYP3A4) and is predominantly excreted via biliary and fecal routes, with approximately 10–20% of the drug eliminated through the kidneys. Colchicine should not be administered in patients with a creatinine clearance rate below 10 mL/min and in those with severe hepatobiliary dysfunction. Colchicine is generally well-tolerated at moderate doses, but higher doses are associated with a significant increase in gastrointestinal side effects, including diarrhea, nausea, and vomiting. Less common adverse effects include myelosuppression, myotoxicity, alopecia, and hepatotoxicity [61,62].

In dermatology, colchicine has been employed in both oral and topical forms for various indications. Although the FDA has approved colchicine solely for managing acute gout attacks and familial Mediterranean fever [57], it has shown efficacy in treating various dermatological conditions, especially those involving neutrophilic inflammation. Nevertheless, the supporting evidence for most of these alternative uses is inconsistent, with the notable absence of RCTs to firmly establish their effectiveness.

### 3.2. Autoimmune Bullous Disorders

The overall evidence on the use of colchicine for autoimmune bullous diseases remains sparse, with varying degrees of success reported across different conditions. In linear IgA disease, a rapid response has been documented in pediatric patients in past studies. Colchicine has also shown promise in MMP, where a retrospective study of 15 patients revealed that 1–1.5 mg/day, used as an adjunct to oral prednisolone, resulted in complete resolution in five patients and improvement in three others [63]. Additionally, its use in EBA and dermatitis herpetiformis was explored, though gastrointestinal side effects were common, necessitating dose reduction. While colchicine appears to have potential in these conditions, further research is needed to establish its efficacy and safety across the spectrum of autoimmune bullous diseases.

### 3.3. Neutrophilic Dermatoses

Colchicine is regarded as a viable treatment option for the treatment of Sweet syndrome according to a case report documenting the resolution of lesions with 0.5 mg of colchicine thrice daily by 1 week and to a retrospective analysis of 20 patients showing that 18 out of 20 patients experienced rapid symptom relief within 2–5 days on a daily dose of 1–1.5 mg [64].

Colchicine has been documented as an effective treatment for PG in small case series and case reports, particularly when other therapies have proven ineffective. Four documented cases of PG (two idiopathic and two associated with inflammatory bowel disease) demonstrated that a daily dose of 1–2 mg colchicine resulted in the resolution of lesions. However, the lack of prospective studies and randomized controlled trials (RCTs) leaves its efficacy unconfirmed.

Colchicine may be considered an appropriate option for managing vasculitis, although available evidence provides mixed results. Colchicine has been used to treat cutaneous vasculitis, though its efficacy remains debated. The effect of the drug has been reported in LCV and in necrotizing vasculitis, but a randomized controlled trial by Sais et al. did not find significant benefits of colchicine over topical emollients for LCV, and a multicentric trial (ARAMIS) is currently ongoing to evaluate and compare the efficacy of azathioprine, colchicine, and DDS in the treatment of isolated skin vasculitis [65].

Colchicine may be particularly useful in difficult-to-treat forms of psoriasis, such as pustular psoriasis and palmoplantar pustulosis. In treating PPP, the use of colchicine with daily doses of 1–2 mg resulted in partial or complete remission in 31 of 32 patients within 2–8 weeks. Additional studies have shown varying degrees of efficacy: in a double-blind, placebo-controlled trial, Thestrup-Pedersen et al. observed significant improvement with 0.5 mg of colchicine administered three times daily over four weeks compared to a placebo. Despite these findings, a 2006 Cochrane review highlighted that while colchicine can be modestly beneficial, its use is often limited by a high incidence of gastrointestinal side effects [66]. Zachariae et al. observed complete remission within two weeks in three out of four patients with pustular psoriasis treated with colchicine.

Available evidence suggests that colchicine, particularly when used in combination with antibacterial agents, may represent an effective therapeutic option for the management of HS. Colchicine has been used in the treatment of HS with mixed results. A retrospective study by Liakou AI et al., which included 44 HS patients, demonstrated that colchicine, either as monotherapy or in combination with doxycycline, significantly improved both clinical severity and quality of life [67]. Similarly, a prospective open-label study treated 20 patients with a regimen of 100 mg of oral minocycline once daily and 0.5 mg of colchicine twice daily for 6 months, followed by a 3-month maintenance phase with colchicine alone. All patients showed clinical improvement after 3 months, with further progress observed over time [68]. However, conflicting findings exist, such as the study by van der Zee et al., which found no clinical improvement in 8 HS patients treated with 0.5 mg of colchicine twice daily for up to 4 months, despite notable gastrointestinal side effects [69].

Colchicine has demonstrated effectiveness in both Behçet’s syndrome and recurrent aphthous stomatitis (RAS). For Behçet’s syndrome, a double-blind clinical trial showed that 1.5 mg of colchicine daily significantly reduced erythema nodosum and arthralgias. In a clinical trial by Masuda et al., colchicine was found to be as effective as cyclosporine in symptom control, making it a safer and more affordable option for Behçet’s syndrome. In RAS, a study by Katz et al. reported that colchicine at 0.5 mg administered three times daily led to a marked reduction in pain and ulcer count, with a 77% pain reduction and 71% decrease in ulcers [70]. Colchicine is regarded as a potent anti-inflammatory treatment effective for both RAS and Behçet’s syndrome.

## 4. Tetracyclines

Tetracyclines are a class of broad-spectrum bacteriostatic antibiotics, mainly composed of tetracycline, doxycycline, lymecycline and minocycline. Their class is named after the common molecular structure consisting of a basic four-hydrocarbon ring. Tetracyclines inhibit bacterial protein synthesis by binding to the 30S ribosomal subunit. Their broad spectrum and high tolerability have kept them in use for many infections and they are also commonly used in non-infective diseases for their anti-inflammatory properties. Moreover, new modified-release (sub-antimicrobial) preparations of doxycycline and minocycline appear to minimize the development of antibiotic resistance without losing anti-inflammatory effects.

### 4.1. Mechanism of Action

Tetracyclines passively diffuse through bacterial membrane pores. Their bacteriostatic potential is based on ribosome inactivation: when bound to the 30S subunit, tetracyclines hinder the binding of the aminoacyl-tRNA to the acceptor site on the mRNA–ribosome complex and inhibit protein biosynthesis. The biological activity of tetracyclines is strongly dependent on metal chelation, influencing the binding of bacterial ribosomes and metalloproteinase-family enzymes and mediating the mechanism of resistance. Calcium (Ca^2+^) is the main cation found in plasma, while magnesium (Mg^2+^) is found in intercellular space. Chelation affects drug absorption in the gastrointestinal tract, necessitating an interval between tetracycline administration and meals. Mechanisms underlying the anti-inflammatory activity of tetracyclines are summarized in Table 2.

### 4.2. Autoimmune Bullous Disorders

Doxycycline and minocycline have been used for their ability to inhibit matrix metalloproteinases (MMPs) as enzymes that contribute to tissue degradation and blister formation in these diseases. By modulating immune responses and reducing inflammation, tetracyclines may help improve disease outcomes and reduce the need for more aggressive immunosuppressive treatments, therefore offering a safer long-term option despite limited evidence of efficacy. The BLISTER study demonstrated that initiating treatment with doxycycline (200 mg/day) achieves comparable short-term blister control and superior long-term safety compared to prednisolone at 0.5 mg/kg/day [71]. A systematic review and meta-analysis confirmed the efficacy and safety of tetracyclines, with an average treatment duration of 3.7 months, a positive response rate of 81.3%, a recurrence rate of 28.3%, and 41.9% reporting side effects. Tetracyclines have also been used as steroid-sparing agents during the consolidation phase following corticosteroid therapy. Although tetracyclines are less effective than systemic corticosteroids, no significant differences were found in recurrence rates [72,73].

### 4.3. Neutrophilic Dermatoses

Tetracyclines have shown efficacy in PG by helping to control inflammation, acting as immunosuppressive agents, and promoting ulcer healing. If no response is observed within six to eight weeks, treatment should be discontinued [74]. Doxycycline is favored due to its anti-inflammatory properties, favorable side-effect profile, and its ability to reduce the need for systemic immunosuppressants, although most reports highlight the benefit of minocycline. The efficacy data for minocycline are primarily based on case reports where it has been used successfully as monotherapy or as a glucocorticoid-sparing agent [75,76]. Minocycline is considered a valid option for limited PG, where it may be used for monotherapy after an insufficient response to initial treatments. While efficacy data are limited, its relatively favorable side-effect profile supports its use before proceeding to more aggressive immunosuppressive therapies. Tetracyclines are also effective as adjunct therapies, such as in paraneoplastic PG, where minocycline, combined with low-dose steroids, offers an approach that minimizes the risks of immunosuppression in patients with metastatic malignancy [77]. Minocycline has also been reported to successfully treat atypical bullous PG and vegetative PG, especially in mild, non-aggressive variants and over a shorter duration [75]. Farrel et al. reported a case of PG in the penis, resistant to treatment with high doses of prednisolone, in which the addition of minocycline achieved remission, supporting minocycline as a valid strategy for difficult locations [78].

Systemic steroid therapy, colchicine, and dapsone are typically employed in the treatment of LCV; however, tetracyclines represent an effective therapeutic option due to their mechanism of action inhibiting neutrophilic chemiotaxis and phagocytosis, which play a critical role in the pathogenesis of this condition [6,79].

In erythema, elevatum diutinum tetracyclines are currently used successfully, and side effects are minimal, even in prolonged treatments. [80].

The action of tetracyclines on neutrophil chemotaxis provides a rationale for their use in pustular psoriasis, but efficacy data are currently sparse. Barlow and Schulz reported that two out of six female patients with a variant of generalized pustular psoriasis showed a positive response to tetracycline therapy [81].

Tetracyclines are largely used in the clinical management of moderate forms of HS: they offer dual benefits in HS treatment, acting both as antibiotics and anti-inflammatory agents. However, a small randomized controlled trial compared topical clindamycin at 0.1% b.i.d. with tetracycline 500 mg b.i.d. in a double-blind, double-dummy study: the trial found no significant differences in physicians’ or patients’ assessment of overall effect, soreness, nodules, or abscesses [82].

The management of Behçet’s disease is complex and requires a multidisciplinary approach, with treatment plans tailored to the severity of the condition and the specific organs affected. While tetracyclines could potentially be helpful due to their anti-inflammatory properties, they have not yet been widely reported in the literature as a treatment option for Behçet’s disease [83,84].

### 4.4. Eosinophilic Dermatoses

Tetracyclines are effective in these conditions due to their ability to inhibit the production of eosinophil-activating cytokines and reduce eosinophil chemotaxis.

In the case of Wells syndrome reported by Stam-Westerveld et al., minocycline at 100 mg/day was used in combination with niacinamide, following a protocol derived from the treatment of bullous pemphigoid. This regimen reduced the frequency and severity of eruptions. [85] However, despite this treatment, multiple relapses occurred during the follow-up perio.

### 4.5. Interface Dermatitis

Tetracyclines have been employed in these conditions due to their ability to modulate immune response and reduce inflammation.

The anti-inflammatory properties of tetracycline antibiotics have been investigated in LP, with recent evidence suggesting that their mechanism may involve the inhibition of T cell response. However, no studies to date have conclusively established the efficacy of tetracyclines for treating cutaneous LP. One case report demonstrated the efficacy of topical tetracycline for erosive oral LP, while another reported improvement in LP pemphigoides following treatment with tetracycline and nicotinamide [86]. In an open-label pilot study conducted by Hantash et al., the response of cutaneous LP to tetracyclines was disappointing. Only 1 out of 13 patients treated with tetracycline or doxycycline achieved a complete response, which may be attributed to the natural disease course, where spontaneous remission occurs over months to years [87]. For 15 patients, doxycycline was chosen as the initial treatment due to its relatively low adverse effect profile and lack of required laboratory monitoring. Of these patients, 27% (n = 4) showed improvement. Clinically, it was suggested that a stepwise approach be taken, with doxycycline offered as a first-line systemic agent [88].

In a case series involving 29 patients with lichen planopilaris, the Cleveland Clinic Foundation found that tetracycline at 1 g/day was the only treatment to show statistically significant improvement. Both tetracycline and topical or intralesional steroids elicited good responses from patients [89].

### 4.6. Granulomatous Diseases

Tetracyclines have been shown to inhibit granuloma formation in vitro, and minocycline has demonstrated the ability to inhibit T cell proliferation in vitro, while incyclinide, a chemically modified doxycycline lacking antibiotic activity but possessing enhanced anti-inflammatory properties, has shown potential in suppressing MMPs and may represent a promising treatment for granulomatous diseases [90]. Marcus et al. reported six cases of treatment-resistant GA that resolved after three months of therapy with a combination of rifampin at 600 mg, ofloxacin at 400 mg, and minocycline hydrochloride at 100 mg administered monthly. The complete clearance of the plaque occurred within 3 to 5 months, though some patients experienced post-inflammatory hyperpigmentation [91]. Duarte et al. described a case of multi-resistant GGA that showed near-complete resolution after a 10-week course of doxycycline at 100 mg/day, with sustained improvement for over a year [92]. The use of tetracyclines in multi-resistant GA is noteworthy and may offer a low-toxicity alternative to first-line topical corticosteroid therapy.

Tetracycline derivatives, due to their favorable safety profile, have emerged as valuable treatments for extensive cutaneous sarcoidosis and may be combined with topical and intralesional corticosteroids. Although data are mainly derived from open, nonrandomized trials, minocycline has shown promising results, achieving complete responses in up to two-thirds of cases [93]. Tetracyclines help manage the extensive cutaneous manifestations of sarcoidosis through their anti-inflammatory and immunomodulatory effects, particularly on T cells. Some evidence supports the involvement of anti-infectious mechanisms in tetracycline’s efficacy in sarcoidosis. Recent studies have identified DNA specific to mycobacteria and *Propionibacterium acne* in sarcoidal lymph nodes, with the latter being known for its sensitivity to tetracyclines. In a retrospective study by Steen et al., minocycline appeared to be a promising option for treating moderate cutaneous sarcoidosis, with 74% of patients (20 out of 27) responding to therapy. Based on the current literature, the authors recommend a stepwise therapeutic approach, with minocycline as the first-line therapy, followed by hydroxychloroquine, methotrexate, thalidomide, and biological agents [94]. Tetracyclines may represent an important therapeutic tool even in cases of sarcoidosis affecting special sites. One case involved a 45-year-old woman with a 7-year history of subcutaneous nodular sarcoidosis and new-onset dyspnea. Although initially treated with corticosteroids, her disease recurred upon withdrawal. A 6-month course of doxycycline at 200 mg/day led to complete remission, underscoring the importance of systemic evaluation and regular monitoring in such cases [95]. Similarly, in tattoo-associated sarcoidosis, a patient achieved the complete clearance of skin lesions and normalization of chest X-ray findings after 4 months of treatment with a mid-potency topical steroid and doxycycline [96].

The immunomodulatory and anti-infectious properties of minocycline have been suggested to explain its efficacy in inflammatory diseases beyond sarcoidosis, such as rheumatoid arthritis. Tetracyclines should also be considered an alternative therapy in cases that show the resistance of sarcoidal lesions to antimalarial drugs [97].

Tetracyclines have been used to treat ulcerated NL, often in combination with topical corticosteroids or tacrolimus, to promote ulcer healing. Mahé et al. reported a case where doxycycline significantly improved ulcerated NL, likely due to its anti-infective, anti-inflammatory, and immunomodulatory properties [81].

## 5. Conclusions

Anti-neutrophilic drugs have established themselves as versatile agents in the treatment of a wide range of dermatological conditions. Their anti-inflammatory and immunomodulatory properties make them particularly valuable in managing autoimmune bullous diseases, neutrophilic dermatoses, eosinophilic dermatoses, interface dermatitis, and granulomatous diseases. By inhibiting inflammatory pathways, reducing cytokine production, and modulating immune responses, they contribute significantly to the treatment and management of these complex skin conditions. Their use continues to evolve as our understanding of these diseases deepens, and they remain a cornerstone of dermatological therapy.

Finally, an emerging approach for refractory skin conditions mediated by activated neutrophils is provided by granulocyte and monocyte adsorption apheresis (GMA). This non-farmacologic strategy is based on the extracorporeal removal of activated granulocytes using a cellulose acetate beads column, which selectively traps these cells. GMA is a promising alternative in patients who have failed conventional therapies for generalized pustular psoriasis, pyoderma gangrenosum, Behçet’s disease, and hidradenitis suppurativa. The strengths of GMA lie in its favorable tolerability and peculiar mode of action that is able to deplete inflammation without causing immunodeficiency. Additional data from rigorous studies are needed to optimize treatment regimens and support a potential role for GMA in specific immune-mediated inflammatory skin disorders.

## Figures and Tables

**Table 1 biomolecules-14-01515-t001:** Uses and mechanisms of action of anti-neutrophilic drugs for dermatological and autoimmune disorders.

Treatment	Use	Mechanism of Action
Dapsone	Dermatitis herpetiformis Autoimmune bullous disorders Sweet syndrome and pyoderma gangrenosum Erythema elevatum diutinum Urticarial vasculitis Leukocytoclastic vasculitis Pustular psoriasis Hidradenitis suppurativa Eosinophilic dermatoses: wells syndrome, granuloma faciale, eosinophilic annular erythema and chronic idiopathic urticaria Cutaneous lupus erythematosus Recurrent erythema multiforme Generalized granuloma annulare Rheumatoid papular eruption	Inhibits neutrophil chemotaxis. Reduces oxidative damage in tissues. Reduces tumor necrosis factor.
Colchicine	Autoimmune bullous disorders Sweet syndrome and pyoderma gangrenosum Pustular psoriasis Hidradenitis suppurativa Behçet’s syndrome	Disrupts microtubule formation. Reduces neutrophil migration and activity. Decreases inflammatory cytokines.
Tetracyclines	Acne Rosacea Autoimmune bullous disorders Sweet syndrome and pyoderma gangrenosum Erythema elevatum diutinum Leukocytoclastic vasculitis Hidradenitis suppurativa Pustular psoriasis Wells syndrome Lichen planus pilaris Generalized granuloma annulare and cutaneous sarcoidosis	Inhibits neutrophil chemotaxis. Reduces cytokines. Reduces matrix metalloproteinases. Prevents oxidative damage. Anti-apoptotic effect.

**Table 2 biomolecules-14-01515-t002:** Effects and mechanisms of action of anti-neutrophilic drugs.

Treatment	Effect	Mechanism
		Modulation of neutrophil chemotaxis
Dapsone	Anti-neutrophilic properties	Inhibition of myeloperoxidase reducing oxidative stress
		Amelioration of neutrophil-mediated damage
		Inhibition of neutrophil chemotaxis
Colchicine	Anti-neutrophilic properties	Inhibition of neutrophil degranulation
		Modulation of the inflammatory cascade
	Inhibition of MMPs	Binding zinc or calcium ions in the enzyme structure
Tetracycline	Anti-inflammatory activity	Inhibition of bacterial breakdown products, Reduced production of pro-inflammatory cytokines (IL-1β, IL-8, TNF), Inhibition of leukocyte migration, Reduced NO by inhibiting iNOS activity
	Antioxidant effect	ROS scavenging and preventing oxidative damage to cell structures through the phenolic ring
	Anti-apoptotic effect	Decreased caspase expression, Altered mitochondrial membrane potential, Inhibition of cytochrome C release, affecting the intrinsic pathway of apoptosis

Abbreviations: iNOS, inducible nitric oxide synthase; MMP, metalloproteinases; NO, nitric oxide; ROS, reactive oxygen species.

## Data Availability

No new data were created or analyzed in this study. Data sharing is not applicable to this article.

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
