# Peer review of "Repurposing Historic Drugs for Neutrophil-Mediated Inflammation in Skin Disorders"

_biomolecules, 2024, doi:10.3390/biom14121515_

Round 1
Reviewer 1 Report
Comments and Suggestions for Authors
This is an interesting well written review covering extensively the topic.
I have only some minor comments:
There are too many references and some of them are pretty old.
It would be nice and helpful if the authors provide a table summarizing the treatments (drugs) their use and possibly the mechanism of action.
Author Response
Reviewer 1
This is an interesting well written review covering extensively the topic.
I have only some minor comments:
Dear Reviewer,
Thank you for your careful review of our manuscript and for your encouraging comments.
We have considered your suggestions, and our responses are provided below:
There are too many references and some of them are pretty old.
The total number of references has been reduced from 153 to a total of 97; citations to older works have been removed, except those reporting clinical evidence that may still be useful.
It would be nice and helpful if the authors provide a table summarizing the treatments (drugs) their use and possibly the mechanism of action.
A relevant table has been added. Please see: “Table 1. Uses and mechanisms of action of anti-neutrophilic drugs for dermatological and autoimmune disorders.”
Reviewer 2 Report
Comments and Suggestions for Authors
This is a review described about neutrophil mediated inflammation in relation to various skin diseases. A few therapeutic agents (Dapsone, Colchicine and Tetracyclines) and their anti-neutrophilic mechanism of action was discussed, and the anti-inflammatory, immunomodulatory properties, efficacy of the agents were discussed with the support from scientific studies reported in the literature.
Comments:
1. The introduction is a bit confusing. Some of the skin diseases authors gave a very brief discussion on it, some in detail. I suggest the authors to give a more equal discussion on the skin diseases.
2. In Introduction, authors can some summarise them into a table form.
3. Some of the short forms spotted, i.e. hasn't
4. There is no good flow of section in 2.3. The authors need to improve the flow of the discussion. Currently there subsection of 2.3 seems unrelated to each other. And the scope of discussion for each disease is also inconsistent.
5. Propionibacterium acnes need to be italicised.
6. Similar comment to no. 1, I suggest the authors to relook into the discussion of each section, as the focus given in each discussion is not balanced. Some were discussed briefly, some were discussed in very detailed.
Author Response
Reviewer 2
This is a review described about neutrophil mediated inflammation in relation to various skin diseases. A few therapeutic agents (Dapsone, Colchicine and Tetracyclines) and their anti-neutrophilic mechanism of action was discussed, and the anti-inflammatory, immunomodulatory properties, efficacy of the agents were discussed with the support from scientific studies reported in the literature.
Dear Reviewer,
Thank you for your efforts on our manuscript and for your constructive comments.
We have carefully considered our comments and made the necessary changes in the text. Please find our responses point by point below:
Comments:
- The introduction is a bit confusing. Some of the skin diseases authors gave a very brief discussion on it, some in detail. I suggest the authors to give a more equal discussion on the skin diseases.
The introduction paragraph has been carefully revised to provide a better balance between paragraphs. Specifically, those paragraphs that were unnecessarily long have been shortened (see, for example, 1.3.3 and 1.3.4). Moreover, a brief sentence on treatments has been added for each disorder.
- In Introduction, authors can some summarise them into a table form.
A table has been added to summarize the uses and mechanisms of the drugs. Please see “Table 1. Uses and mechanisms of action of anti-neutrophilic drugs for dermatological and autoimmune disorders.”
- Some of the short forms spotted, i.e. hasn't
We apologize for these writing mistakes, that have been corrected.
- There is no good flow of section in 2.3. The authors need to improve the flow of the discussion. Currently there subsection of 2.3 seems unrelated to each other. And the scope of discussion for each disease is also inconsistent.
Section 2.3 has been reorganized and rewritten to improve clarity and cohesion. Separation into subsections has been removed here and in 3.3 and 4.3.
- Propionibacterium acnes need to be italicised.
Thank you, names of species are now written in italics.
- Similar comment to no. 1, I suggest the authors to relook into the discussion of each section, as the focus given in each discussion is not balanced. Some were discussed briefly, some were discussed in very detailed.
Thank you for your suggestion; paragraphs that were excessively lengthy have been carefully revised to improve balance between parts and overall clarity.
Round 2
Reviewer 2 Report
Comments and Suggestions for Authors
Suggestions:
1. Remove section 1.3.2 write-up. Too brief and not much information given which did not show its importance.
2. Table 2, 3, and 4 can consider to combine as one table.
Author Response
[Comment 1.] "Remove section 1.3.2 write-up. Too brief and not much information given which did not show its importance."
[Response] Thank you for your suggestion. We have removed section 1.3.2 write up.
[Comment 2.] "Table 2, 3, and 4 can consider to combine as one table."
[Response] As kindly suggested, tables 2,3, and 4 have been merged into a single table.